# 3D Wavelet Finite-Element Modeling of Frequency-Domain Airborne EM Data Based on B-Spline Wavelet on the Interval Using Potentials

Lingqi Gao [1], Changchun Yin [1,*], Ning Wang [1], Jiao Zhu [1], Yunhe Liu [1], Xiuyan Ren [1], Bo Zhang [1] and Bin Xiong [2]

[1] College of Geo-Exploration Sciences and Technology, Jilin University, Changchun 130026, China; gaolq19@mails.jlu.edu.cn (L.G.); ningwang19@mails.jlu.edu.cn (N.W.); zhujiao18@mails.jlu.edu.cn (J.Z.); liuyunhe@jlu.edu.cn (Y.L.); renxiuyan@jlu.edu.cn (X.R.); zhang2019bo@jlu.edu.cn (B.Z.)

[2] College of Earth Sciences, Guilin University of Technology, Guilin 541006, China; xiongbin@glut.edu.cn

* Correspondence: yinchangchun@jlu.edu.cn

**Abstract:** We present a wavelet finite-element method (WFEM) based on B-spline wavelets on the interval (BSWI) for three-dimensional (3D) frequency-domain airborne EM modeling using a secondary coupled-potential formulation. The BSWI, which is constructed on the interval (0, 1) by joining piecewise B-spline polynomials between nodes together, has proved to have excellent numerical properties of multiresolution and sparsity and thus is utilized as the basis function in our WFEM. Compared to conventional basis functions, the BSWI is able to provide higher interpolating accuracy and boundary stability. Furthermore, due to the sparsity of the wavelet, the coefficient matrix obtained by BSWI-based WFEM is sparser than that formed by general FEM, which can lead to shorter solution time for the linear equations system. To verify the accuracy and efficiency of our proposed method, we ran numerical experiments on a half-space model and a layered model and compared the results with one-dimensional (1D) semi-analytic solutions and those obtained from conventional FEM. We then studied a synthetic 3D model using different meshes and BSWI basis at different scales. The results show that our method depends less on the mesh, and the accuracy can be improved by both mesh refinement and scale enhancement.

**Keywords:** frequency-domain airborne EM; 3D modeling; coupled-potential formulation; wavelet finite-element method; B-spline wavelet on the interval



## 1. Introduction

Airborne electromagnetic method (AEM) is an efficient and low-cost tool for geophysical exploration. It is very suitable for explorations in areas with complex terrains such as mountains, deserts, swamps and forest-covered areas. Its applications cover mineral explorations, oil and gas, groundwater detections, environmental and engineering investigations, geological disaster monitoring, etc. [1–7]. In recent years, the computer science and numerical simulation algorithms have made remarkable progress, hence, 3D forward modeling and inversions has attracted more and more attention. However, efficient and accurate modeling algorithms are still a great challenge. The existing algorithms for frequency-domain airborne EM modeling includes integral equation (IE), finite-difference (FD), and finite-element (FE) methods [8–12]. The IE method is one of the earliest numerical methods used in 3D EM forward modeling. It needs only to discretize the anomalous areas. Therefore, the coefficient matrix is normally smaller, which is beneficial to memory consumption and a fast solution. However, the coefficient matrix is asymmetric and dense, which makes the IE method limited to simple models. The FD method is normally constructed on hexahedral staggered grids [13]. It is simple in theory and easy to implement. However, most applications based on the FD method are still limited to structured grids, although scientists have also paid attention to the unstructured grid. As a result, this method

depends heavily on the mesh for improving the accuracy, especially for complex models. The FE method divides the model into a series of non-overlapping meshes and uses basis functions to approximate the unknown field variation in each element. Compared to other numerical methods, the FE method has received more attention in geophysical EM forward modeling in the last decades. This is mainly because the FE method has more flexibility and higher accuracy, as one can work on unstructured meshes for complicated models such as terrain.

However, there exist disadvantages with the conventional FE method, in that the modeling accuracy depends heavily on the mesh [14,15]. One needs to refine the mesh for higher accuracy, which can lead to the rapid growth of unknowns, especially for the hexahedral mesh that will be discussed in this paper. High-order FEM can improve the accuracy by enhancing the order of interpolation basis functions [15,16], but the matrix will become much denser when higher-order basis functions are used, and the time consumption rises dramatically. Additionally, strong oscillations are likely to occur in the results when the order of polynomial interpolation is high. This is called the Runge phenomenon [17]. In this paper, we introduce the WFEM to ameliorate the deficiencies in our airborne EM modeling.

Wavelet is a useful mathematical tool to approximate functions. Different from the common interpolation based on polynomials, the wavelet theory constructs a multiresolution framework. By combining the wavelets at different scales together, a given function can theoretically be approximated at any resolution. The initial research on wavelet theory began in many different disciplines in the 20th century [18–23]. Subsequently, more and more attention has been paid to the excellent properties of wavelets. Currently, it has been widely applied to image processing, signal processing and many other fields [24,25]. As for geophysics, the applications of wavelet cover geophysical inversions [26–28], modeling of seismic wave propagation [29], geophysical data processing [30], and so on. Meanwhile, in the field of numerical simulation, the wavelet-related algorithms have also been developed for years, such as wavelet finite-difference, wavelet finite-volume, and wavelet finite-element methods [31,32]. Among these methods, the WFEM has gained more preference and developed fast. The WFEM uses wavelet functions or scaling functions as the basis function in order to obtain a nested multi-level solution with high accuracy and efficiency. Among different wavelet basis, the DB wavelet is the most commonly used one in WFEM [22]. Sarkar and Adve (1994) applied WFEM to solve the Maxwell's equations for two-dimensional (2D) waveguide problems [33]; Castro and Barbosa (2006) introduced the DB wavelet for 2D structure analysis using WFEM [34]. In geophysics, Zhang et al. (2005) proposed a DB-based WFEM for solving 2D wave equation in the fluid-saturated porous media [35]; Feng et al. (2016) applied the DB-based WFEM to solve the 2D wave equation for ground penetrating radar (GPR) [36]; Hussain et al. (2016, 2017) introduced a WFEM using DB wavelet to solve 1D and 2D marine controlled-source EM forward problem [37,38]; Chen and Li (2019) proposed a WFEM for 3D marine controlled-source electromagnetic modeling in anisotropic medium using DB wavelet [39]. Additionally, the WFEM based on other wavelet bases such as polynomial wavelet and second-generation wavelet has also made remarkable progress [40,41], which we do not address in detail here.

However, there exists important factors that prevent the DB wavelet from continuously developing. One such factor is that the DB wavelet does not have explicit expressions. Thus, it is a challenge to calculate its integration or derivative. The second is that the DB wavelet often needs to be truncated in WFEM because it is defined on the whole real axis. Numerical instability is hard to avoid if one uses the truncated DB wavelet to approximate the unknown functions in a finite range. These two factors affect both the applicability and accuracy of the DB wavelet in WFEM. In this paper, we select the wavelet basis, called the B-spline wavelet on the interval (BSWI) that was first proposed by Chui and Quak [42]. Its construction is derived from the B-spline function. The B-spline function is defined by piecewise polynomials between nodes [43]. The B-spline interpolation has proved to have better characteristics than polynomial interpolating because they are able to avoid the Runge phenomenon as the order increases. Meanwhile, the low-order B-spline

interpolation is likely to produce the same effect as high-order polynomial interpolation. Based on the B-spline functions, the BSWI is established by combining B-spline functions between nodes on the interval (0, 1). It inherits all the excellent properties of wavelet and B-spline function. Compared to the DB wavelet, the BSWI is more suitable for WFEM for two reasons. Firstly, the BSWI basis has explicit expression. Thus, its integration and derivative can be easily executed. Secondly, the BSWI uses special functions at the boundary to approximate the field near the boundary so that no truncation is needed. Goswami et al. (1995) applied the BSWI to numerical computation for the first time to make the matrix in IE sparser [44]; Xiang et al. (2006) proposed a BSWI-based WFEM to solve plane elastomechanics and moderately thick plate problems [45]; Shen et al. (2020) used the BSWI for FEM to study the 2D elastic wave propagation [46]; Feng et al. (2019) proposed a BSWI-based WFEM for 2D GPR modeling to study the EM responses of porous media, which is so far the only research on BSWI-based FEM in geophysics [47].

Normally, the EM forward problems can be solved in terms of the electric or magnetic field or the coupled vector-scalar potentials [48,49]. For the first case, the nodal-based FEM is not able to guarantee the normal discontinuity and tangential continuity of the electric field at the surfaces [50]. Furthermore, it fails to satisfy the zero-divergence condition. Therefore, spurious modes may exist. Instead, the edge-based finite-element method proposed by Nédélec (1980) uses vector basis functions defined along the edges to approximate the electric field [50]. This guarantees the continuity of the tangential field component while allowing the discontinuity of the normal component at the same time. Moreover, it can avoid the spurious modes because the construction of vector basis functions satisfies the divergence-free condition of the electric field in a source-free mesh. Thus, the edge-based FEM has obtained more applications in EM modeling. For the second case, the magnetic vector potential and the electric scalar potential are in nature continuous at surfaces, thus, the nodal-based FEM is able to solve it properly. In order to assure the uniqueness of the solution, a Coulomb or Lorentz gauge needs to be applied [51]. In this paper, we enforce a Coulomb gauge to avoid the spurious modes as well as to form a symmetric coefficient matrix [52]. The coupled-potential formulation has been applied to the modeling of global geomagnetic induction, MT, and the controlled-source EM [53–56]. In this paper, we present a secondary coupled-potential formulation and their WFEM solutions. The paper is organized as follows. We first derive the governing equations of 3D frequency-domain AEM forward modeling based on coupled potentials. After introducing the theory of BSWI and BSWI-based WFEM, we illustrate how to construct and solve the linear system using WFEM. We then verify the accuracy and demonstrate the advantages of our method via 3D numerical experiments.

## 2. Methodology

### 2.1. Governing Equations

In this paper, the frequency-domain forward problem is solved in terms of coupled vector-scalar potentials. Assuming a time dependence of $e^{i\omega t}$, the diffusive Maxwell's equation after ignoring the displacement current can be written as

$$\nabla \times \mathbf{E} = -i\omega\mu_0 \mathbf{H} \tag{1}$$

$$\nabla \times \mathbf{H} = \mathbf{J}_s + \sigma \mathbf{E} \tag{2}$$

where $\omega$ is the angular frequency, $\mu_0$ represents the magnetic permeability in the free-space, $\sigma$ is the conductivity, while $\mathbf{J}_s$ denotes the source current distribution. The electric field $\mathbf{E}$ and magnetic field $\mathbf{H}$ can be represented in terms of the magnetic vector potential $\mathbf{A}$ and the electric scalar potential $\Psi$, i.e.,

$$\mathbf{H} = \frac{1}{\mu_0}\nabla \times \mathbf{A} \tag{3}$$

$$\mathbf{E} = -i\omega(\mathbf{A} + \nabla\Psi) \tag{4}$$

Inserting Equations (3) and (4) into (1) and (2), we obtain the following curl-curl equation for the potentials:

$$\nabla \times \nabla \times \mathbf{A} + i\omega\mu_0\sigma(\mathbf{A} + \nabla\Psi) = \mu_0\mathbf{J}_s \tag{5}$$

Equation (5) is likely to cause spurious modes, because the divergence of the vector potential is not uniquely defined. To guarantee the uniqueness of the solution to the vector potential $\mathbf{A}$ and to form a symmetric matrix, we introduce the Coulomb gauge $\nabla \cdot \mathbf{A} = 0$ into Equation (5). Expanding the first term in Equation (5) and taking into account the vector identity $\nabla \times \nabla \times \mathbf{A} - \nabla(\nabla \cdot \mathbf{A}) = -\nabla^2\mathbf{A}$ and the Coulomb gauge, we can get the following equation:

$$\nabla^2\mathbf{A} - i\omega\mu_0\sigma(\mathbf{A} + \nabla\Psi) = -\mu_0\mathbf{J}_s \tag{6a}$$

To satisfy the divergence-free condition of the current density $\nabla \cdot \mathbf{J} = 0$, we add an auxiliary equation in our model solutions, namely

$$\nabla \cdot [i\omega\mu_0\sigma(\mathbf{A} + \nabla\Psi)] = -\nabla \cdot (\mu_0\mathbf{J}_s) \tag{6b}$$

In the following, we utilize a secondary potential formulation to avoid the singularities near the source. The secondary potentials are defined by

$$\mathbf{A} = \mathbf{A}_p + \mathbf{A}_s \tag{7a}$$

$$\Psi = \Psi_p + \Psi_s \tag{7b}$$

where $\mathbf{A}_p$, $\Psi_p$ are, respectively, the primary vector potential and scalar potential, while $\mathbf{A}_s$ and $\Psi_s$ are the secondary potentials. Substituting Equation (7a,b) into Equation (6a,b), we get

$$\nabla^2\mathbf{A}_s - i\omega\mu_0\sigma(\mathbf{A}_s + \nabla\Psi_s) = -i\omega\mu_0\Delta\sigma(\mathbf{A}_p + \nabla\Psi_p) \tag{8a}$$

$$\nabla \cdot [i\omega\mu_0\sigma(\mathbf{A}_s + \nabla\Psi_s)] = \nabla \cdot [i\omega\mu_0\Delta\sigma(\mathbf{A}_p + \nabla\Psi_p)] \tag{8b}$$

where $\Delta\sigma = \sigma - \sigma_p$. From Equation (4) it is seen that the primary potentials $(\mathbf{A}_p, \Psi_p)$ can be replaced by the primary electric field $\mathbf{E}_p$. Thus, Equation (8a,b) can be simplified as

$$\nabla^2\mathbf{A}_s - i\omega\mu_0\sigma(\mathbf{A}_s + \nabla\Psi_s) = -\mu_0\Delta\sigma\mathbf{E}_p \tag{9a}$$

$$\nabla \cdot [i\omega\mu_0\sigma(\mathbf{A}_s + \nabla\Psi_s)] = \nabla \cdot (\mu_0\Delta\sigma\mathbf{E}_p) \tag{9b}$$

where $\mathbf{E}_p$ denotes the responses of a dipole in a free-space.

Since the EM fields are very small at a great distance from the source, we impose a Dirichlet boundary condition at the outer boundary $\Gamma$, i.e.,

$$(\mathbf{A}_s, \Psi_s)_\Gamma = (0, 0) \tag{10}$$

In summary, the governing Equation (9a,b) and the boundary condition of Equation (10) together constitute the coupled-potential system for solving frequency-domain airborne EM forward problems.

### 2.2. B-Spline Wavelet on the Interval

The wavelet provides a multiresolution framework for representing functions. In general, if a function $w(x)$ satisfies $\int_{-\infty}^{\infty} w(x)dx = 0$, it can be defined as a wavelet. By stretching and translation, a set of functions can be created from $w(x)$, i.e.,

$$w_{j,k}(x) = 2^{\frac{j}{2}}w\left(2^j x - k\right) \tag{11}$$

where $j, k \in Z$. Note that this set of functions can form a basis on $L^2(R)$. Any function $f(x)$ from $L^2(R)$ can be represented as

$$f(x) = \sum_j \sum_k d_{j,k} w_{j,k}(x) \tag{12}$$

where $d_{j,k}$ denotes the wavelet coefficients. In this way, the wavelet basis at each scale can construct a wavelet space $W_j$, and $L^2(R)$ is the direct sum of all these wavelet spaces. This can be simply represented as

$$\boldsymbol{W}_j := clos_{L^2(R)} \left\{ w_{j,k}(x) : k \in Z \right\}, j \in Z \tag{13}$$

$$L^2(R) = \sum_{j \in Z} \boldsymbol{W}_j = \cdots + \boldsymbol{W}_{-1} + \boldsymbol{W}_0 + \boldsymbol{W}_1 + \cdots \tag{14}$$

where $clos\{\}$ denotes the closure operator. More specifically, $w_{j_0,k}(x)$ for a given $j_0$ is a set that contains all wavelets at scale $j_0$. These wavelets together can form the enclosure space $W_{j_0}$. Thus, any function in this space can be represented as a linear combination of the wavelets in $W_{j_0}$. Furthermore, all the functions in $L^2(R)$ can then be represented as a linear combination of wavelets in all wavelet spaces.

From Equation (13), we can further construct the space $V_j$ by

$$V_j = \cdots + W_{j-2} + W_{j-1}, j \in Z \tag{15}$$

It is clear that

$$V_j = V_{j-1} + W_{j-1} \tag{16}$$

Equations (15) and (16) show that $\{V_j\}$ forms a multiresolution nested space (see Figure 1) that satisfies the following relationship:

$$\{0\} \subset \cdots \subset V_{-1} \subset V_0 \subset V_1 \subset \cdots \subset L^2(R) \tag{17}$$

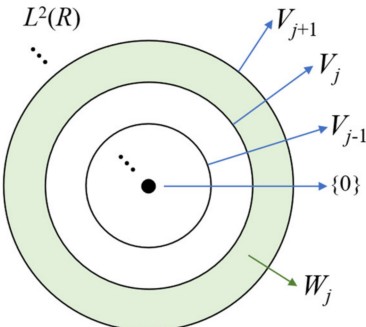

**Figure 1.** Multiresolution nested space.

According to wavelet theory, similar to wavelet functions and wavelet spaces, there also exist scaling functions $s_{j,k}(x)$ that can construct space $V_j$, so that any function can be decomposed, namely

$$f(x) = \sum_k^{j=j_0} c_{j_0,k} s_{j_0,k}(x) \tag{18}$$

Furthermore, Equation (18) can also be written as

$$
\begin{aligned}
f(x) &= \sum_{k}^{j=j_0-1} c_{j,k} s_{j,k}(x) + \sum_{k}^{j=j_0-1} d_{j,k} w_{j,k}(x) \\
&= \sum_{k}^{j=j_0-Z} c_{j,k} s_{j,k}(x) + \sum_{j=j_0-Z}^{j=j_0-1} \sum_{k} d_{j,k} w_{j,k}(x)
\end{aligned}
\tag{19}
$$

where $Z$ can be any positive integer. In this way, a function on $L^2(R)$ can be represented by scaling functions and wavelet functions at any resolution. More specifically, scaling functions approximate the general information, while wavelet functions describe details at certain scales. In this paper, we utilize scaling functions as the basis function in our finite-element method for two reasons. The first is that BSWI scaling functions can accurately hold the general information at a given scale, which means that the unknown field inside each element can be well approximated. The second is that the space $\{V_j\}$ constructed by scaling functions is a set of nested multiresolution spaces, so that it is natural and direct for us to enhance the scale of the basis functions and we can approximate the unknown field at a detailed resolution.

The B-spline wavelet on the interval is normally defined on a knot sequence on the reference interval $(0, 1)$ and any function $f(x)$ on the interval $(a, b)$ can be mapped to the interval $(0, 1)$ by a simple transformation formula $\xi = (x - a)/(b - a)$. In general, for a given scale $j$ ($j \in Z^+$) and order $m$, the interval $(0, 1)$ is divided into $2^j$ segments, then $m-1$ knots are added to each endpoint as virtual multi-knots to describe the information at the boundary. In this way, the knot sequence $\left\{\xi_k^j\right\}_{k=m+1}^{2^j+m-1}$ with a total number of $2^j + 2m-1$ is obtained. B-spline polynomials are then formed between these knots. Subsequently, BSWI can be constructed by joining these B-spline polynomials together. Note that to avoid confusing BSWI with the electric scalar potential $\Psi$, we use $s$ and $w$ in this paper to represent the scaling functions and wavelet functions of BSWI, respectively.

The BSWI basis at different scales can be transformed into each other because they satisfy a recursive relationship [42]. Taken as an example, Goswami et al. (1995) gave the expressions of the BSWI basis at 0th scale and $m$th order [44]. Thus, the $j$th scale and $m$th order of BSWI (simply denoted as $\text{BSWI}_{mj}$) scaling functions $s_{m,k}^j(\xi)$ and the corresponding wavelet functions $w_{m,k}^j(\xi)$ can be evaluated by the following equations:

$$
s_{m,k}^j(\xi) = \begin{cases}
s_{m,k}^l\left(2^{j-l}\xi\right), & k = -m+1,\dots,-1 \ (0-\text{boundary}) \\
s_{m,2^j-m-k}^l\left(1 - 2^{j-l}\xi\right), & k = 2^j - m + 1,\dots,2^j - 1 \ (1-\text{boundary}) \\
s_{m,0}^l\left(2^{j-l}\xi - 2^{-l}k\right), & k = 0,\dots,2^j - m \ (\text{inner})
\end{cases}
\tag{20a}
$$

$$
w_{m,k}^j(\xi) = \begin{cases}
w_{m,k}^l\left(2^{j-l}\xi\right), & k = -m+1,\dots,-1 \ (0-\text{boundary}) \\
w_{m,2^j-2m-k+1}^l\left(1 - 2^{j-l}\xi\right), & k = 2^j - m + 2,\dots,2^j - m \ (1-\text{boundary}) \\
w_{m,0}^l\left(2^{j-l}\xi - 2^{-l}k\right), & k = 0,\dots,2^j - 2m + 1 \ (\text{inner})
\end{cases}
\tag{20b}
$$

In this paper, we set $j = 1$ to be the initial scale. The scaling functions and wavelet functions at all scales can be calculated according to Equation (20a,b). There are $m-1$ 0-boundary and 1-boundary scaling functions and wavelet functions, and $2^j-m + 1$ inner scaling functions and $2^j-2m + 2$ wavelet functions (only wavelet functions at the 1st scale do not satisfy the index in Equation (20b), because they do not contain any inner wavelet functions). After we have obtained the $\text{BSWI}_{mj}$ basis, the scaling functions and the wavelet functions on $(0, 1)$ can be written in vector format as

$$
\boldsymbol{s} = \left\{ s_{m,-m+1}^j(\xi), s_{m,-m+2}^j(\xi), \cdots, s_{m,2^j-1}^j(\xi) \right\}
\tag{21a}
$$

$$w = \left\{ w^j_{m,-m+1}(\xi), w^j_{m,-m+2}(\xi), \cdots, w^j_{m,2^j-m}(\xi) \right\} \tag{21b}$$

It must be pointed out that the boundary BSWI basis are different from the inner ones, so that the truncated boundary can be properly described. Meanwhile, the BSWI basis has local support, which means that each scaling function or wavelet function can only affect a part of the interval (0, 1). The advantage is that the boundary can be well approximated no matter how high the scale increases. On the contrary, the conventional FEM interpolation that uses polynomials defined on the whole interval can cause a strong oscillation as the order of polynomials keeps increasing. Thus, our method can avoid the so-called Runge phenomenon.

Figure 2 shows the linear (i.e., $m = 2$) scaling functions and the wavelet functions at 1st scale and 2nd scale. We use the linear scaling functions in this paper as the basis functions in WFEM to approximate the unknown fields and to make the coefficient matrix sparse. We define $n = 2^j + m - 1$, which means the number of BSWI nodes for a given order and scale.

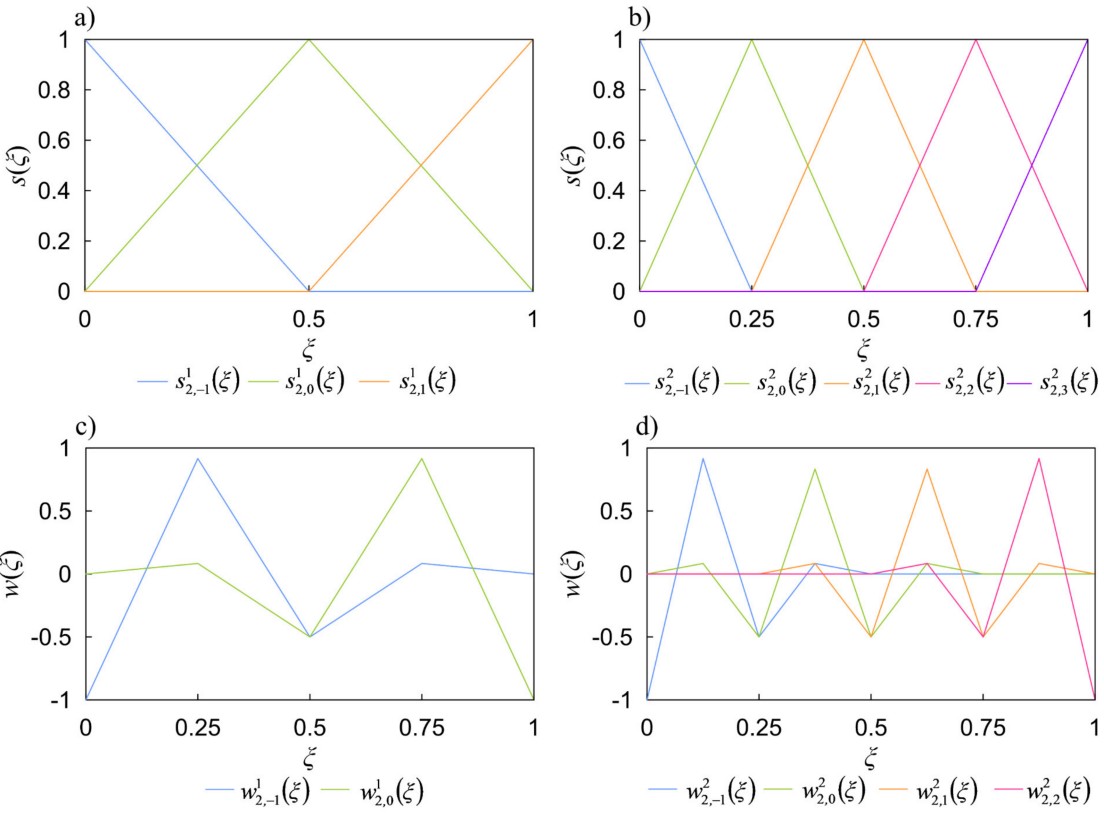

**Figure 2.** Linear ($m = 2$) scaling functions and wavelet functions of BSWI: (**a**) scaling functions at scale $j = 1$; (**b**) scaling functions at scale $j = 2$; (**c**) wavelet functions at scale $j = 1$; (**d**) wavelet functions at scale $j = 2$.

Tensor product is a simple and direct way to construct BSWI basis on higher dimensions. Higher-dimensional BSWI basis naturally inherits all the characteristics from 1D BSWI basis. Moreover, this strategy can also form a nested multiresolution framework on higher dimension as the scale $j$ varies (same as that shown in Figure 1). The scaling function $s$ and the tensor product subspace $F_j$ for 3D cases can be written as

$$s = s_1 \otimes s_2 \otimes s_3 \tag{22a}$$

$$F_j = V^1_j \otimes V^2_j \otimes V^3_j \tag{22b}$$

where $s_1 = \left\{ s_{m,-m+1}^j(\xi), s_{m,-m+2}^j(\xi), \cdots, s_{m,2^j-1}^j(\xi) \right\}$ denotes the vector combined by scaling functions on $\xi$ axis, while $s_2$ and $s_3$ are defined on $\eta$ and $\zeta$ axis, and $\otimes$ denotes the Kronecker symbol.

Based on this discussion, 2D and 3D scaling functions are shown in Figures 3 and 4. To avoid confusion, we renumbered them from 1 to $n^2$ for 2D case and from 1 to $n^3$ for 3D case. Figure 3 shows the 2D BSWI scaling functions at the corner (Figure 3a,d), inside (Figure 3b,e), and on the edge (Figure 3c,e) at scale $j = 1$ and $j = 2$. Figure 4 shows the 3D BSWI scaling functions at the corner (Figure 4a,e), on the edge (Figure 4b,f), on the surface (Figure 4c,g) and inside (Figure 4d,h) at scale $j = 1$ and $j = 2$.

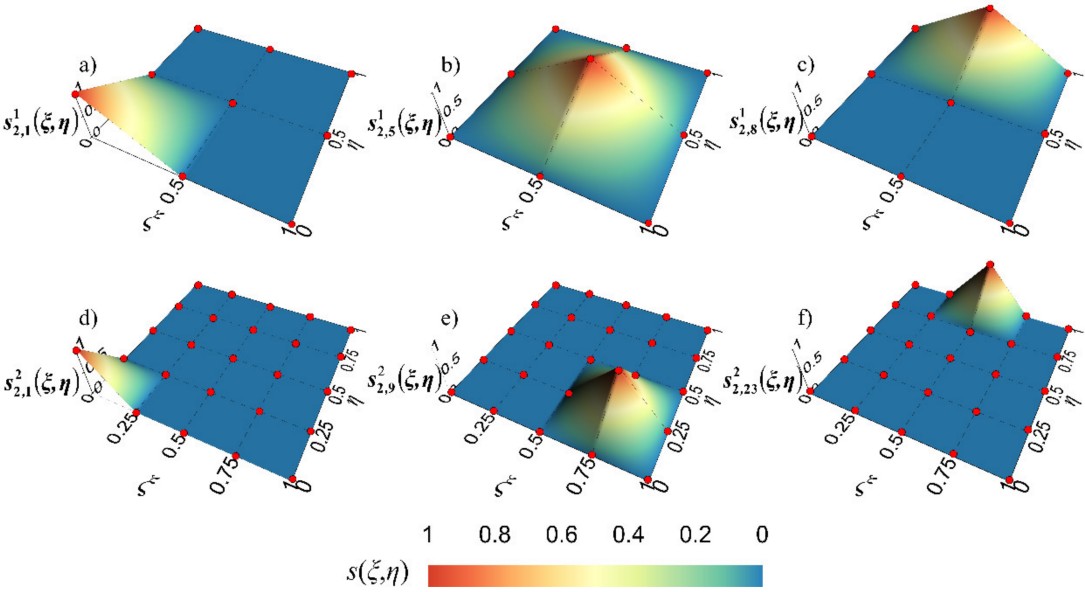

**Figure 3.** 2D linear BSWI basis at 1st scale (the upper row) and 2nd scale (the under row): (**a**) $s_{2,1}^1(\xi,\eta)$ ; (**b**) $s_{2,5}^1(\xi,\eta)$; (**c**) $s_{2,8}^1(\xi,\eta)$; (**d**) $s_{2,1}^2(\xi,\eta)$; (**e**) $s_{2,9}^2(\xi,\eta)$; (**f**) $s_{2,23}^2(\xi,\eta)$.

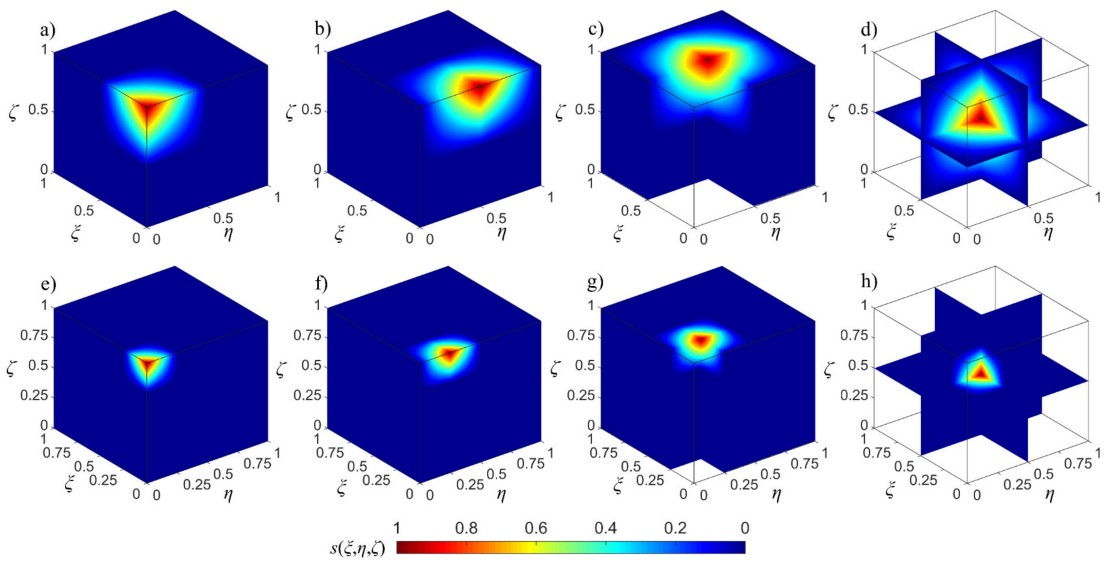

**Figure 4.** 3D linear BSWI basis at 1st scale (the upper row) and 2nd scale (the under row): (**a**) $s_{2,19}^1(\xi,\eta,\zeta)$; (**b**) $s_{2,22}^1(\xi,\eta,\zeta)$; (**c**) $s_{2,23}^1(\xi,\eta,\zeta)$; (**d**) $s_{2,14}^1(\xi,\eta,\zeta)$; (**e**) $s_{2,101}^2(\xi,\eta,\zeta)$; (**f**) $s_{2,106}^2(\xi,\eta,\zeta)$; (**g**) $s_{2,107}^2(\xi,\eta,\zeta)$; (**h**) $s_{2,63}^2(\xi,\eta,\zeta)$.

### 2.3. BSWI Based Wavelet Finite-Element Method

In traditional finite-element method, the basis functions (also known as shape functions) are normally used to describe the unknown field function $u(x, y, z)$ in each element, i.e.,

$$u(x, y, z) = N u^e \tag{23}$$

where $N$ denotes the basis function vector, while $u^e$ is a column vector that denotes the physical degree of freedom (DOF).

As for the WFEM, according to Equation (18), any unknown function in a given hexahedral element can be mapped into a reference mesh and approximated by 3D scaling functions at scale $j$, namely

$$u(x, y, z) \rightarrow u(\xi, \eta, \zeta) = s a^e \tag{24}$$

where $u(\xi, \eta, \zeta)$ represents the unknown function in the reference hexahedral domain, while $a^e = \{a_1, a_2, \cdots, a_n, \cdots, a_{n^2}, \cdots, a_{n^3}\}^{\mathrm{T}}$ denotes the wavelet coefficients column vector. In order to convert the modeling problem from wavelet domain to physical domain, which can be understood as a bridge between Equations (23) and (24), we introduce the following transformation matrix:

$$\mathbf{T} = \mathbf{R}^{-1} \tag{25}$$

$$\mathbf{R} = \mathbf{R}_1 \otimes \mathbf{R}_2 \otimes \mathbf{R}_3 \tag{26}$$

where $\mathbf{T}$ is the transformation matrix, $\mathbf{R}$ can be calculated from the following equations:

$$\begin{cases} \mathbf{R}_1 = \left\{ s_1^{\mathrm{T}}(\xi_1), s_1^{\mathrm{T}}(\xi_2), \cdots, s_1^{\mathrm{T}}(\xi_n) \right\}^{\mathrm{T}} \\ \mathbf{R}_2 = \left\{ s_2^{\mathrm{T}}(\xi_1), s_2^{\mathrm{T}}(\xi_2), \cdots, s_2^{\mathrm{T}}(\xi_n) \right\}^{\mathrm{T}} \\ \mathbf{R}_3 = \left\{ s_3^{\mathrm{T}}(\xi_1), s_3^{\mathrm{T}}(\xi_2), \cdots, s_3^{\mathrm{T}}(\xi_n) \right\}^{\mathrm{T}} \end{cases} \tag{27}$$

Note that the transformation matrix describes the value of scaling functions on each node in the reference mesh that can be simply taken as a wavelet transform. Thus, we have

$$a^e = \mathbf{T} u^e \tag{28}$$

The above procedure constructs the connection between the DOFs in physical space and the wavelet coefficients in the wavelet space. Equation (24) can be further expressed as

$$u(x, y, z) \rightarrow u(\xi, \eta, \zeta) = s \mathbf{T} u^e = N u^e \tag{29}$$

where $N = s\mathbf{T}$ is the basis function vector in WFEM.

So far, we have established the basis function of the BSWI-based WFEM method. The following parts of WFEM are similar to conventional FEM, and will be discussed it in the next section.

Figure 5a shows the layout and corresponding local index of nodes in the reference mesh for scale $j$ and order $m$, while Figure 5b,c shows the BSWI$_{21}$ and BSWI$_{22}$ elements. Note that the layout of nodes is the same as that in the 3D BSWI basis, however, the basis functions are not exactly the same as the 3D BSWI basis functions because of the transformation matrix.

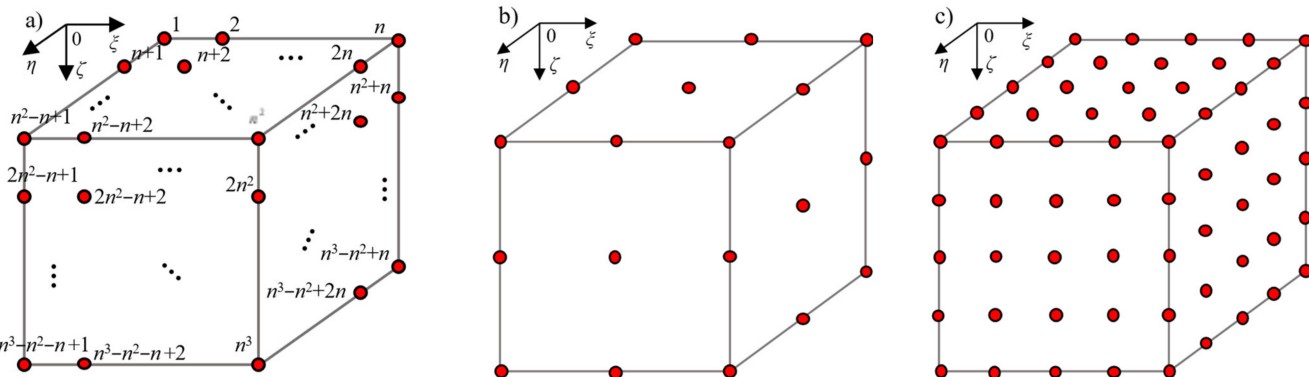

**Figure 5.** Layout of 3D BSWI elements: (**a**) BSWI$_{mj}$ element; (**b**) BSWI$_{21}$ element; (**c**) BSWI$_{22}$ element.

### 2.4. WFEM Analysis

In this section, we expand the vector potential **A** in the governing Equation (9a,b) into $x$, $y$, $z$ components and obtain the following equations:

$$\nabla^2 \mathbf{A}_{sx} - i\omega\mu_0\sigma\left(\mathbf{A}_{sx} + \frac{\partial\Psi_s}{\partial x}\right) = -\mu_0\Delta\sigma\mathbf{E}_{px} \tag{30a}$$

$$\nabla^2 \mathbf{A}_{sy} - i\omega\mu_0\sigma\left(\mathbf{A}_{sy} + \frac{\partial\Psi_s}{\partial x}\right) = -\mu_0\Delta\sigma\mathbf{E}_{py} \tag{30b}$$

$$\nabla^2 \mathbf{A}_{sz} - i\omega\mu_0\sigma\left(\mathbf{A}_{sz} + \frac{\partial\Psi_s}{\partial x}\right) = -\mu_0\Delta\sigma\mathbf{E}_{pz} \tag{30c}$$

$$i\omega\mu_0\sigma\left[\left(\frac{\partial\mathbf{A}_{sx}}{\partial x} + \frac{\partial\mathbf{A}_{sy}}{\partial y} + \frac{\partial\mathbf{A}_{sz}}{\partial z}\right) + \nabla\cdot(\nabla\Psi_s)\right] = -\nabla\cdot(\mu_0\Delta\sigma\mathbf{E}_p) \tag{30d}$$

and we discretize by presenting them in weak format as

$$-\left(\nabla\mathbf{N}^{\mathrm{T}}, \nabla\mathbf{A}_{sx}\right)_\Omega - i\omega\mu_0\sigma\left(\mathbf{N}^{\mathrm{T}}, \mathbf{A}_{sx} + \frac{\partial\Psi_s}{\partial x}\right)_\Omega = -\mu_0\Delta\sigma\left(\mathbf{N}^{\mathrm{T}}, \mathbf{E}_{px}\right)_\Omega \tag{31a}$$

$$-\left(\nabla\mathbf{N}^{\mathrm{T}}, \nabla\mathbf{A}_{sy}\right)_\Omega - i\omega\mu_0\sigma\left(\mathbf{N}^{\mathrm{T}}, \mathbf{A}_{sy} + \frac{\partial\Psi_s}{\partial y}\right)_\Omega = -\mu_0\Delta\sigma\left(\mathbf{N}^{\mathrm{T}}, \mathbf{E}_{py}\right)_\Omega \tag{31b}$$

$$-\left(\nabla\mathbf{N}^{\mathrm{T}}, \nabla\mathbf{A}_{sz}\right)_\Omega - i\omega\mu_0\sigma\left(\mathbf{N}^{\mathrm{T}}, \mathbf{A}_{sz} + \frac{\partial\Psi_s}{\partial z}\right)_\Omega = -\mu_0\Delta\sigma\left(\mathbf{N}^{\mathrm{T}}, \mathbf{E}_{pz}\right)_\Omega \tag{31c}$$

$$-i\omega\mu_0\sigma\left(\nabla\mathbf{N}^{\mathrm{T}}, \mathbf{A}_s\right)_\Omega - i\omega\mu_0\sigma\left(\nabla\mathbf{N}^{\mathrm{T}}, \nabla\Psi_s\right)_\Omega = -\mu_0\Delta\sigma\left(\mathbf{N}^{\mathrm{T}}, \nabla\cdot\mathbf{E}_p\right)_\Omega \tag{31d}$$

where $(u, v)_\Omega = \int_\Omega uvd\Omega$. Then, we use the basis function of WFEM in Equations (24) and (29) to present the unknown potentials. For each mesh containing $n^3$ nodes, the weak format can be represented as a linear equation $\mathbf{K}^e\mathbf{u}^e = \mathbf{b}^e$, where $\mathbf{K}^e$ is a complex symmetric matrix:

$$\mathbf{K}^e = \begin{pmatrix} \mathbf{K}^{11} & 0 & 0 & \mathbf{K}^{14} \\ 0 & \mathbf{K}^{22} & 0 & \mathbf{K}^{24} \\ 0 & 0 & \mathbf{K}^{33} & \mathbf{K}^{34} \\ \mathbf{K}^{41} & \mathbf{K}^{42} & \mathbf{K}^{43} & \mathbf{K}^{44} \end{pmatrix} \tag{32}$$

where the elements can be written as

$$\mathbf{K}^{11} = \mathbf{K}^{22} = \mathbf{K}^{33} = \iiint_\Omega \left[-\left(\frac{\partial\mathbf{N}^{\mathrm{T}}}{\partial x}\frac{\partial\mathbf{N}}{\partial x} + \frac{\partial\mathbf{N}^{\mathrm{T}}}{\partial y}\frac{\partial\mathbf{N}}{\partial y} + \frac{\partial\mathbf{N}^{\mathrm{T}}}{\partial z}\frac{\partial\mathbf{N}}{\partial z}\right) - i\omega\mu_0\sigma\mathbf{N}^{\mathrm{T}}\mathbf{N}\right]d\Omega \tag{33a}$$

$$\mathbf{K}^{44} = -i\omega\mu_0\sigma \iiint\limits_{\Omega} \left( \frac{\partial \mathbf{N}^{\mathrm{T}}}{\partial x} \frac{\partial \mathbf{N}}{\partial x} + \frac{\partial \mathbf{N}^{\mathrm{T}}}{\partial y} \frac{\partial \mathbf{N}}{\partial y} + \frac{\partial \mathbf{N}^{\mathrm{T}}}{\partial z} \frac{\partial \mathbf{N}}{\partial z} \right) d\Omega \tag{33b}$$

$$\mathbf{K}^{14} = -i\omega\mu_0\sigma \iiint\limits_{\Omega} \left( \mathbf{N}^{\mathrm{T}} \frac{\partial \mathbf{N}}{\partial x} \right) d\Omega \tag{33c}$$

$$\mathbf{K}^{24} = -i\omega\mu_0\sigma \iiint\limits_{\Omega} \left( \mathbf{N}^{\mathrm{T}} \frac{\partial \mathbf{N}}{\partial y} \right) d\Omega \tag{33d}$$

$$\mathbf{K}^{34} = -i\omega\mu_0\sigma \iiint\limits_{\Omega} \left( \mathbf{N}^{\mathrm{T}} \frac{\partial \mathbf{N}}{\partial z} \right) d\Omega \tag{33e}$$

$$\mathbf{K}^{41} = -i\omega\mu_0\sigma \iiint\limits_{\Omega} \left( \frac{\partial \mathbf{N}^{\mathrm{T}}}{\partial x} \mathbf{N} \right) d\Omega \tag{33f}$$

$$\mathbf{K}^{42} = -i\omega\mu_0\sigma \iiint\limits_{\Omega} \left( \frac{\partial \mathbf{N}^{\mathrm{T}}}{\partial y} \mathbf{N} \right) d\Omega \tag{33g}$$

$$\mathbf{K}^{43} = -i\omega\mu_0\sigma \iiint\limits_{\Omega} \left( \frac{\partial \mathbf{N}^{\mathrm{T}}}{\partial z} \mathbf{N} \right) d\Omega \tag{33h}$$

while the right-hand side can be written as

$$\boldsymbol{b}^e = -\mu_0 \Delta\sigma \left( \begin{array}{c} \left( \mathbf{N}^{\mathrm{T}}, \mathbf{N} \right)_{\Omega} \mathbf{E}_{px}, \left( \mathbf{N}^{\mathrm{T}}, \mathbf{N} \right)_{\Omega} \mathbf{E}_{py}, \left( \mathbf{N}^{\mathrm{T}}, \mathbf{N} \right)_{\Omega} \mathbf{E}_{pz}, \\ \left( \frac{\partial \mathbf{N}^{\mathrm{T}}}{\partial x}, \mathbf{N} \right)_{\Omega} \mathbf{E}_{px} + \left( \frac{\partial \mathbf{N}^{\mathrm{T}}}{\partial y}, \mathbf{N} \right)_{\Omega} \mathbf{E}_{py} + \left( \frac{\partial \mathbf{N}^{\mathrm{T}}}{\partial z}, \mathbf{N} \right)_{\Omega} \mathbf{E}_{pz} \end{array} \right)^{\mathrm{T}} \tag{34}$$

where $\mathbf{E}_{px}$, $\mathbf{E}_{py}$, $\mathbf{E}_{pz}$ are the primary electrical field column vector in $x$, $y$, and $z$ directions. As mentioned before, we assume the primary field to be excited by a transmitting source in the free-air space, so we have for a vertical magnetic dipole (VMD) the electrical field:

$$E_{px} = \frac{i\omega\mu_0 m}{4\pi r^2} (ik_0 r + 1) e^{-ik_0 r} \frac{y}{r} \tag{35a}$$

$$E_{py} = -\frac{i\omega\mu_0 m}{4\pi r^2} (ik_0 r + 1) e^{-ik_0 r} \frac{x}{r} \tag{35b}$$

$$E_{pz} = 0 \tag{35c}$$

where $m$ denotes the dipole moment, $k = \sqrt{i\omega\mu_0\sigma + \omega^2\mu_0\varepsilon_0}$, $x, y, z$ are the relative distances between the source and the receiver at the $x$-, $y$-, and $z$- axis, $r = \sqrt{x^2 + y^2 + z^2}$.

In the above equations, the 3D integrations of the basis function can reduce to the tensor product of 1D integrations of the BSWI basis in the reference domain because the scaling functions in different directions are orthogonal to each other. We take the first term in Equation (33a) as example to illustrate this process. In fact, from Equation (33a), we can expand the first term as

$$\begin{aligned} \iiint_{\Omega} \frac{\partial \mathbf{N}^{\mathrm{T}}}{\partial x} \frac{\partial \mathbf{N}}{\partial x} d\Omega &= \frac{l_y l_z}{l_x} \int_0^1 \int_0^1 \int_0^1 \frac{\partial \mathbf{N}^{\mathrm{T}}}{\partial \xi} \frac{\partial \mathbf{N}}{\partial \xi} d\xi d\eta d\zeta = \frac{l_y l_z}{l_x} \mathbf{T}^{\mathrm{T}} \int_0^1 \int_0^1 \int_0^1 \frac{\partial \boldsymbol{s}^{\mathrm{T}}}{\partial \xi} \frac{\partial \boldsymbol{s}}{\partial \xi} d\xi d\eta d\zeta \mathbf{T} \\ &= \frac{l_y l_z}{l_x} \mathbf{T}^{\mathrm{T}} \int_0^1 \int_0^1 \int_0^1 \left( \frac{\partial \boldsymbol{s}_1^{\mathrm{T}}}{\partial \xi} \otimes \boldsymbol{s}_2^{\mathrm{T}} \otimes \boldsymbol{s}_3^{\mathrm{T}} \right) \left( \frac{\partial \boldsymbol{s}_1}{\partial \xi} \otimes \boldsymbol{s}_2 \otimes \boldsymbol{s}_3 \right) d\xi d\eta d\zeta \mathbf{T} \\ &= \frac{l_y l_z}{l_x} \mathbf{T}^{\mathrm{T}} \int_0^1 \left( \frac{\partial \boldsymbol{s}_1^{\mathrm{T}}}{\partial \xi} \frac{\partial \boldsymbol{s}_1}{\partial \xi} \right) d\xi \otimes \int_0^1 (\boldsymbol{s}_2^{\mathrm{T}} \boldsymbol{s}_2) d\eta \otimes \int_0^1 (\boldsymbol{s}_3^{\mathrm{T}} \boldsymbol{s}_3) d\zeta \mathbf{T} \end{aligned} \tag{36a}$$

where $l_x$, $l_y$, $l_z$ are the mesh sizes in physical space at $x$-, $y$-, $z$-axis. Similarly, the other integrations shown in Equation (33a–h) can also be written as

$$\iiint\limits_{\Omega} \mathbf{N}^{\mathrm{T}} \mathbf{N} d\Omega = l_x l_y l_z \mathbf{T}^{\mathrm{T}} \int_0^1 \left( \boldsymbol{s}_1^{\mathrm{T}} \boldsymbol{s}_1 \right) d\xi \otimes \int_0^1 \left( \boldsymbol{s}_2^{\mathrm{T}} \boldsymbol{s}_2 \right) d\eta \otimes \int_0^1 \left( \boldsymbol{s}_3^{\mathrm{T}} \boldsymbol{s}_3 \right) d\zeta \mathbf{T} \tag{36b}$$

$$\iiint_\Omega \left( \mathbf{N}^{\mathrm{T}} \frac{\partial \mathbf{N}}{\partial x} \right) d\Omega = l_y l_z \mathbf{T}^{\mathrm{T}} \int_0^1 \left( s_1^{\mathrm{T}} \frac{s_1}{\partial \xi} \right) d\xi \otimes \int_0^1 \left( s_2^{\mathrm{T}} s_2 \right) d\eta \otimes \int_0^1 \left( s_3^{\mathrm{T}} s_3 \right) d\zeta \mathbf{T} \qquad (36c)$$

The 1D integration of scaling functions, wavelet functions or their derivatives in WFEM is normally called connection coefficients. We use $\boldsymbol{\Lambda}_d^{p,q}$ to represent these connection coefficients, where $p$ and $q$ denote the order of the derivate of the scaling functions, while $d$ represents the axis. In this paper, the following four connection coefficients are used (only those on $\xi$ axis are shown as examples).

$$\boldsymbol{\Lambda}_1^{0,0} = \int_0^1 \left( s_1^{\mathrm{T}} s_1 \right) d\xi \qquad (37a)$$

$$\boldsymbol{\Lambda}_1^{1,1} = \int_0^1 \left( \frac{\partial s_1^{\mathrm{T}}}{\partial \xi} \frac{\partial s_1}{\partial \xi} \right) d\xi \qquad (37b)$$

$$\boldsymbol{\Lambda}_1^{1,0} = \int_0^1 \left( \frac{\partial s_1^{\mathrm{T}}}{\partial \xi} s_1 \right) d\xi \qquad (37c)$$

$$\boldsymbol{\Lambda}^{0,1} = \int_0^1 \left( s_1^{\mathrm{T}} \frac{\partial s_1}{\partial \xi} \right) d\xi \qquad (37d)$$

In much existing research on WFEM that apply wavelets (e.g., the DB wavelet) as the basis functions, the calculation of connection coefficients can be a challenging problem [57,58]. The definition on the whole real axis and the lack of explicit expressions, as mentioned above, are the most important issues. In this paper, due to the explicit expression and the definition on the interval (0, 1) of BSWI, we can easily obtain the explicit solutions of connection coefficients. In Appendix A, we have given the expressions of connection coefficients for $m = 2$, $j = 1$ and $m = 2$, $j = 2$.

Finally, the element matrix and right-hand side for each element is formed and integrated together and then the Dirichlet boundary condition is applied, so that we can get the following global linear system for our forward modeling problem, i.e.,

$$\mathbf{K}\boldsymbol{u} = \boldsymbol{b} \qquad (38)$$

As we have discussed before, one reason why wavelet is frequently utilized in numerical methods is that it can make the matrix sparse, as does the BSWI basis. The BSWI has local support, which means that each scaling function can only affect a limited area in the mesh. This will further create fewer non-zero coefficients in the matrix. Figure 6 shows the sparsity pattern of the element matrix and global matrix formed by BSWI$_{21}$ and BBSWI$_{22}$ elements. From the figures it is seen that the BSWI indeed contributes to a sparser matrix; the non-zero coefficients are concentrated around the diagonal, so that a lower time consumption can be expected. We will discuss this via numerical experiments in next section.

Airborne EM modeling is a multi-source problem because the airborne transmitter keeps moving during the survey. To solve the linear equations system efficiently, we use the direct solver MUMPS [59,60]. This method needs only to do the factorization once and replace the source term at the right-hand side for moving transmitters.

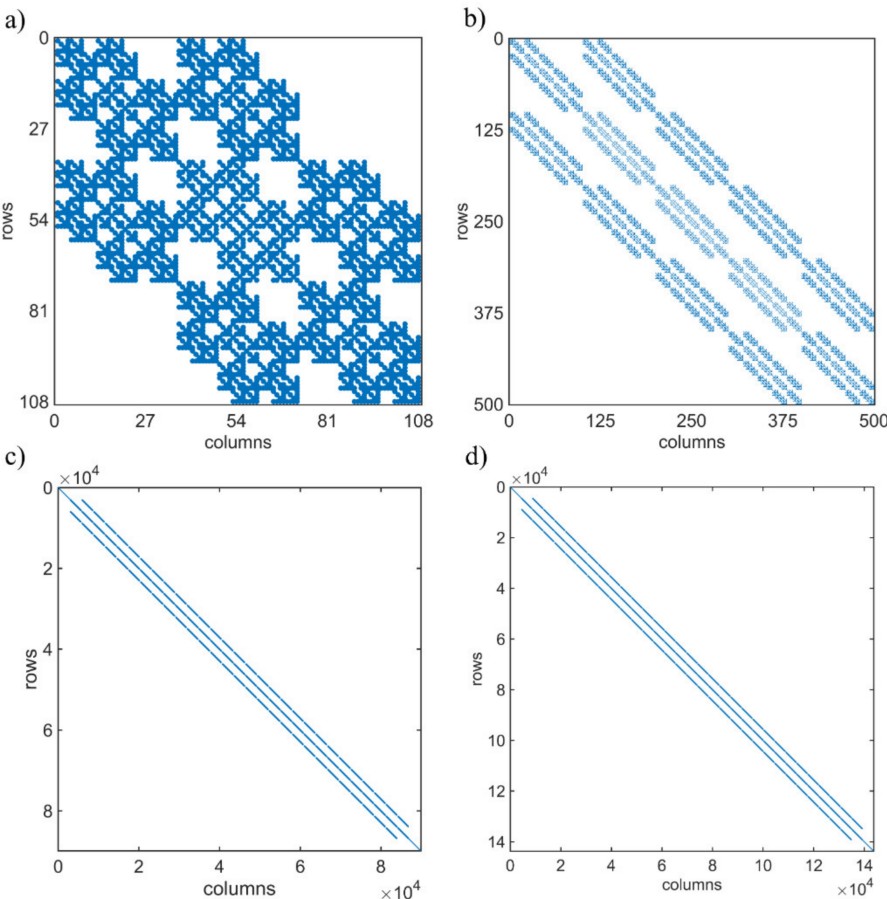

**Figure 6.** Sparsity pattern of the BSWI-based WFEM: (**a**) element matrix of BSWI$_{21}$ elements; (**b**) element matrix of BSWI$_{22}$ elements; (**c**) global matrix when using BSWI$_{21}$ elements; (**d**) global matrix when using BSWI$_{22}$ matrix.

### 2.5. Moving Least-Squares Interpolation

In this section, we apply a moving least-squares interpolation (MLSI) scheme introduced by Tabbara et al. (1994) to recover EM fields $\mathbf{E}_s$ and $\mathbf{H}_s$ from the secondary potentials $\mathbf{A}_s$ and $\Psi_s$ [61]. For this purpose, we assume that each component of the secondary vector potential $\mathbf{A}_s$ and the scalar potential $\Psi_s$ can be approximated by a linear function that can be written as

$$u^* = \boldsymbol{P}\boldsymbol{a} \tag{39}$$

where $u^*$ is the approximate linear function, $\boldsymbol{P} = \{x, y, z, 1\}$ is the variable vector, while $\boldsymbol{a} = \{a_1, a_2, a_3, a_4\}^{\mathrm{T}}$ is the column vector composed of unknown coefficients. The coefficients vector is determined by minimizing the weighted differences between $u^*$ and the computed potential $u$ taking the nearest $N$ nodes to the receiver into account, namely

$$s(a) = \sum_{i=1}^{N} w(\boldsymbol{P}\boldsymbol{a} - u)^2 \tag{40}$$

where $w = e^{-\frac{d^2}{h^2}}$ is the weighted function, $d$ is the distance from the node to the receiver, and $h = max(d)$. $N$ is usually set to be 30~50. The MLSI assures that the derivatives are smooth and accurate so that the EM fields can be well recovered.

## 3. Numerical Experiments

### 3.1. Accuracy Verification

In this section, we first simulate as an example the EM responses of a horizontal coplanar (HCP) AEM system over a homogeneous half-space model to verify the accuracy of our method. The Tx-Rx offset is assumed to be 10m. The flight attitude is 30 m. The earth resistivity is $\rho = 100$ ohm-m. Here, three mesh discretization cases are considered to illustrate how the accuracy can be improved when we refine the mesh or enhance the scale. The model is divided into $10 \times 10 \times 10$, $14 \times 14 \times 14$ and $25 \times 25 \times 25$ elements, labeled as Case 1, Case 2, and Case 3, in which $6 \times 6 \times 6$, $10 \times 10 \times 10$ and $19 \times 19 \times 19$ elements are taken as the calculation domain. Meanwhile, the outer boundary is extended 6000 m in each direction. We calculate 21 logarithmically equal-spaced frequencies ranging from 100 Hz to 215 kHz. The results are compared to the 1D semi-analytic solutions calculated using the code from Yin and Fraser [62]. From Figure 7, it is seen that our method obtains high accuracy with only a small number of meshes. For BSWI$_{21}$ elements in Case 1, the relative errors of both the real and the imaginary part are not able to meet our needs. The maximum relative error can be up to about 15%. This is simply because the mesh discretization is too rough. After the mesh refinement from Case 1 to Case 2, we can see that the relative error shows a sharp reduction, with the relative error reduced to less than 5%. Meanwhile, the time cost is only about 15 s for one frequency (see Table 1). For comparison purposes, we further refine the mesh to Case 3. The result shows that in this case the relative error of the real part becomes less than 1% for most frequencies, while the error of the imaginary part is lower than that of the real part with a maximum value of 1%. This indicates that the results from our method are very accurate. As for the BSWI$_{22}$ elements, the relative error in Case 1 is already under 1% for most frequencies. However, at very high frequencies the error keeps rising to about 8%. However, after the mesh is refined in Case 2, we can obtain similar accuracy as that in Case 3 using BSWI$_{21}$ elements. The maximum relative error of the real part is about 3% at 215 kHz while that of the imaginary part is under 0.6% at all frequencies. This indicates again that the accuracy can be improved when the scale of BSWI basis is increased. From Table 1 it is seen that the BSWI$_{22}$ elements lead to a system with more DOFs than BSWI$_{21}$ elements. As a result, the corresponding time cost becomes higher. Since the results are accurate enough, we do not continue refining the mesh. From this example, we can obviously see that the BSWI-based WFEM depends less on meshes so that one can simulate the EM response with fewer meshes than the conventional FEM. At the same time, as the high-scale functions can describe the unknown field at high resolutions, so we can improve the accuracy by increasing the scale. This provides double tracks for improving the accuracy in our modeling process.

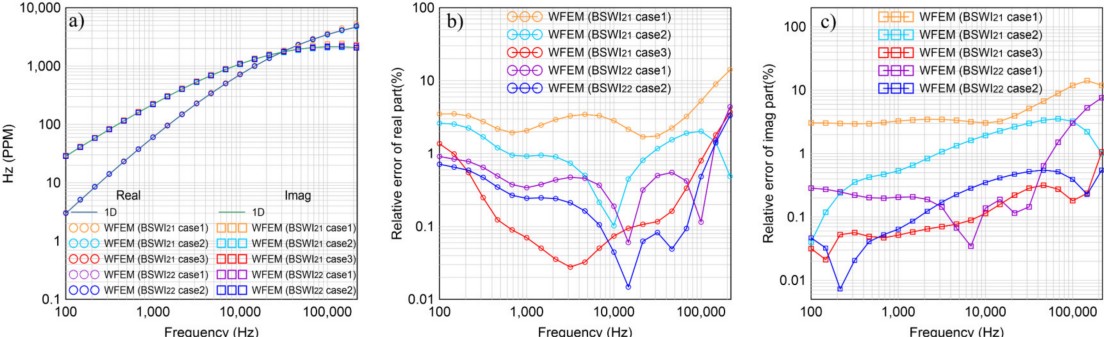

**Figure 7.** Accuracy verification by comparing our WFEM results with semi-analytical solutions for a homogenous half-space model: (**a**) comparison of EM responses; (**b**) relative error of the real part; (**c**) relative error of the imaginary part.

**Table 1.** BSWI basis selection, mesh subdivision, and time consumption of WFEM for a half-space model.

| Type of Elements | Case | Number of Meshes | DOFs | Time Consumption (at 100 Hz) |
|---|---|---|---|---|
| $BSWI_{21}$ | 1 | $10 \times 10 \times 10$ | 37,044 | 4.390 s |
| - | 2 | $14 \times 14 \times 14$ | 97,556 | 15.161 s |
| - | 3 | $25 \times 25 \times 25$ | 530,604 | 340.165 s |
| $BSWI_{22}$ | 1 | $10 \times 10 \times 10$ | 275,684 | 100.281 s |
| - | 2 | $14 \times 14 \times 14$ | 740,772 | 673.973 s |

In the following discussion, we use a layered earth model to further illustrate the efficiency and accuracy of our method. The model is shown in Figure 8. The system parameters are the same as the half-space model in Figure 7. For comparison purposes, we use two methods of the nodal-based FEM and the edge-based FEM [12] (abbreviated as nFEM and eFEM in the following figures, respectively). Here, we use the edge-based FEM to solve the curl-curl equation for the electric field **E** while we use the nodal-based FEM to solve the coupled potentials **A** and $\Psi$. Note that in our comparison the time consumption is taken as the total time, including the time for the factorization and solution. The model used for these three methods is set to be 650 m $\times$ 650 m $\times$ 550 m with the Tx-Rx system located at its center. We use $21 \times 21 \times 24$ $BSWI_{21}$ meshes to do the forward modeling. To make the comparison, two mesh subdivision cases are respectively applied to nodal-based FEM and edge-based FEM. In Case 1, after the mesh subdivision, the DOFs of the three methods are similar (see Table 2). Figure 9 shows the EM responses. From Figure 9b,c one sees that the relative error of WFEM to the semi-analytical solutions is the lowest, which is under 1% at most frequencies for both the real and imaginary parts. Meanwhile, the time consumption is between that of the nodal-based and edge-based FEM. The solutions obtained by the edge-based FEM have the lowest accuracy, the relative error of the imaginary part exceeds 5% at high frequencies. Finally, the nodal-based FEM takes a longer time than our WFEM to provide a solution at low accuracy, which further indicates that WFEM can obtain more accurate results with less time than conventional FEM. In Case 2, we further refine the mesh in Case 1. The details of the mesh subdivision and time consumption are shown in Table 2. From the result comparison and the relative error in Figure 9, one can see that the mesh refinement does improve the accuracy. For edge-based FEM, the accuracy improvement is apparent. However, although the time consumption has been raised to the same level as that of WFEM, the accuracy is still far lower than that of WFEM. As for the nodal-based FEM, the mesh refinement near the system does not contribute much to the accuracy improvement of the real part, but it improves the imaginary part. Thus, we can conclude that the BSWI basis shows better performance than the conventional polynomial interpolation as it captures more local details and the information on the boundaries of meshes. In addition, it contributes to a sparser matrix and thus speeds up the solution of the linear equations. Although the vector basis functions can satisfy the divergence condition $\nabla \cdot \mathbf{E} = 0$ in each element, the edge-based FEM still shows less stability than our nodal-based WFEM using coupled-potential formulation.

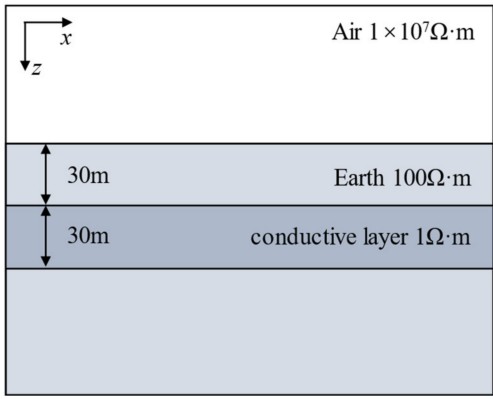

**Figure 8.** A layered earth model.

**Table 2.** Mesh subdivision and time consumption of WFEM, edge-based FEM and nodal-based FEM for the layered earth model in Figure 8.

| Method | Case | Number of Meshes | DOFs | Maximum Mesh Size | Minimum Mesh Size | Time Consumption (at 100 Hz) |
|---|---|---|---|---|---|---|
| WFEM (BSWI$_{21}$) | 1 | 21 × 21 × 24 | 362,404 | 80 m × 80 m × 80 m | 10 m × 10 m × 10 m | 162.929 s |
| Edge-based FEM | 1 | 48 × 48 × 52 | 374,164 | 40 m × 40 m × 40 m | 5 m × 5 m × 5 m | 83.226 s |
| - | 2 | 54 × 54 × 58 | 525,950 | 20 m × 20 m × 20 m | 2.5 m × 2.5 m × 2.5 m | 158.505 s |
| Nodal-based FEM | 1 | 42 × 42 × 48 | 362,404 | 40 m × 40 m × 40 m | 5 m × 5 m × 5 m | 176.186 s |
| - | 2 | 42 × 42 × 52 | 391,988 | 40 m × 40 m × 40 m | 2.5 m × 2.5 m × 2.5 m | 184.530 s |

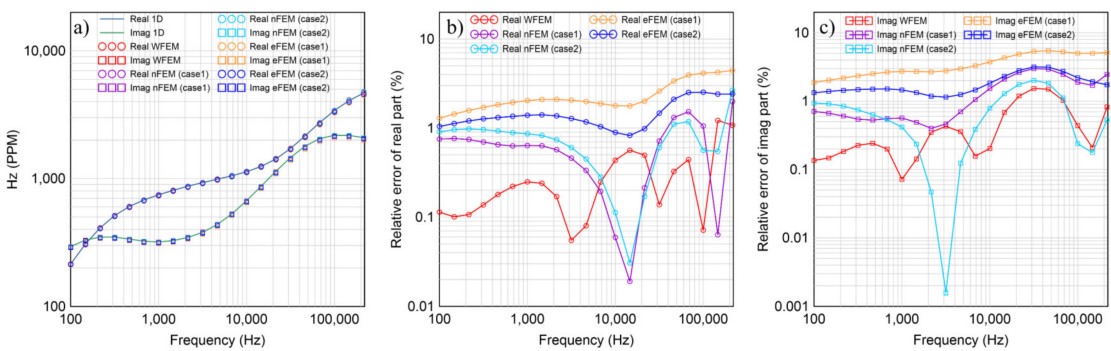

**Figure 9.** Comparison of WFEM, the edge-based and nodal-based FEM for the model in Figure 8: (**a**) comparison of EM responses; (**b**) relative error of the real part; (**c**) relative error of the imaginary part.

### 3.2. 3D Synthetic Models

In this section, we first assume a synthetic Example 1 with a 1 ohm-m conductive body embedded in a 100 ohm-m half-space for our modeling. The size of the conductive body is 80 m × 80 m × 20 m with a top depth of 40 m. The coordinate of its central point is (0 m, 0 m, 50 m). The system parameters are the same as the previous models. We calculate the HCP system responses for a total of 31 Tx-Rx locations along the survey line $y$ = 0 m ranging from $x$ = −150 m to 150 m. The transmitting frequencies are 380, 1600, 6300, and 25,000 Hz, respectively. To illustrate the influence of the BSWI scale and mesh size on the accuracy, we adopt four different subdivisions to discretize the model domain. We use BSWI$_{21}$ elements to divide the model into mesh for Case 1 and 2, while we use BSWI$_{22}$ elements to divide the model for Case 3 and 4. In Case 1, the maximum size of 40 m × 40 m × 40 m is assumed. In Case 2, a mesh refinement is applied to the model without changing the scale of BSWI basis. In Case 3, the scale is increased to $j$ = 2 while the mesh is still the same as in Case 1.

In Case 4, the mesh is refined, and the scale is increased. The EM responses are shown in Figure 10, respectively, while the time consumptions are given in Table 3. For easy comparison, we also present the EM responses calculated by the spectral element method (SEM). According to [63], as a high-order numerical algorithm, SEM has proved to be very accurate. Here, we use the third order of Gauss–Lobatto–Chebyshev (GLC) polynomials as the basis function of SEM for the mesh size of 20 m × 20 m × 20 m. It is seen from Figure 10 that the model in Case 1 failed to obtain reasonable results, as a large difference between Case 1 and SEM can be observed. Table 3 shows the maximum relative errors between our results and the SEM solutions for four cases. It is seen that the maximum relative error between these two methods is 17.4% at 25,000 Hz. This means that the results in Case 1 are not reliable. Meanwhile, the response at high frequencies shows a dramatic oscillation. This is because as the frequency increases, the EM field varies quickly, so that it becomes harder for a rough mesh to simulate. To improve the accuracy, we apply in Case 2 a mesh refinement. Apparently, the results of Case 2 become smoother and closer to the SEM responses, and the relative error drops at all frequencies with a maximum of 4.01%. Similar results can be observed in Case 3, where we enhance the scale of the basis function on the bases of Case 1. Using BSWI22 elements, the relative error falls down when compared to Case 1 and 2 except for the real part of 25,000 Hz (see Figure 11). Among all frequencies, the maximum relative error is 3.43%. Finally, we apply both the mesh refinement and scale enhancement to Case 1. It is seen that the accuracy is largely improved. The maximum relative error between Case 4 and SEM falls to 3.02%. This confirms again that in our new algorithm, we can improve the modeling accuracy by mesh refinement and the scale enhancement.

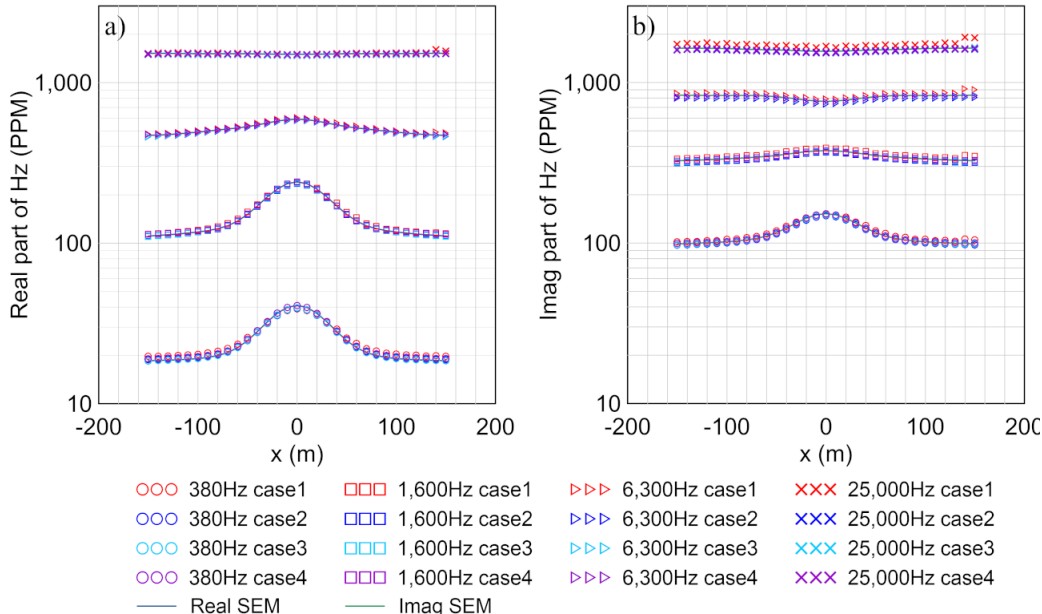

**Figure 10.** EM responses obtained from 4 cases of Example 1 compared to SEM results: (**a**) real part; (**b**) imaginary part.

**Table 3.** Scale of basis function, mesh subdivision, time consumption, and maximum relative error for 4 cases of Example 1.

| Method | Case | Type of Elements | DOFs | Maximum Mesh Size | Minimum Mesh Size | Time Consumption (at 380 Hz) | Maximum Relative Error |
|---|---|---|---|---|---|---|---|
| WFEM | 1 | BSWI$_{21}$ | 89,900 | 40 m × 40 m × 40 m | 40 m × 40 m × 20 m | 27.618 s | 17.4% |
| - | 2 | BSWI$_{21}$ | 414,540 | 20 m × 20 m × 20 m | 20 m × 20 m × 20 m | 310.479 s | 4.01% |
| - | 3 | BSWI$_{22}$ | 792,756 | 40 m × 40 m × 40 m | 40 m × 40 m × 20 m | 811.019 s | 3.43% |
| - | 4 | BSWI$_{22}$ | 968,436 | 20 m × 40 m × 40 m | 20 m × 20 m × 20 m | 1281.788 s | 3.02% |

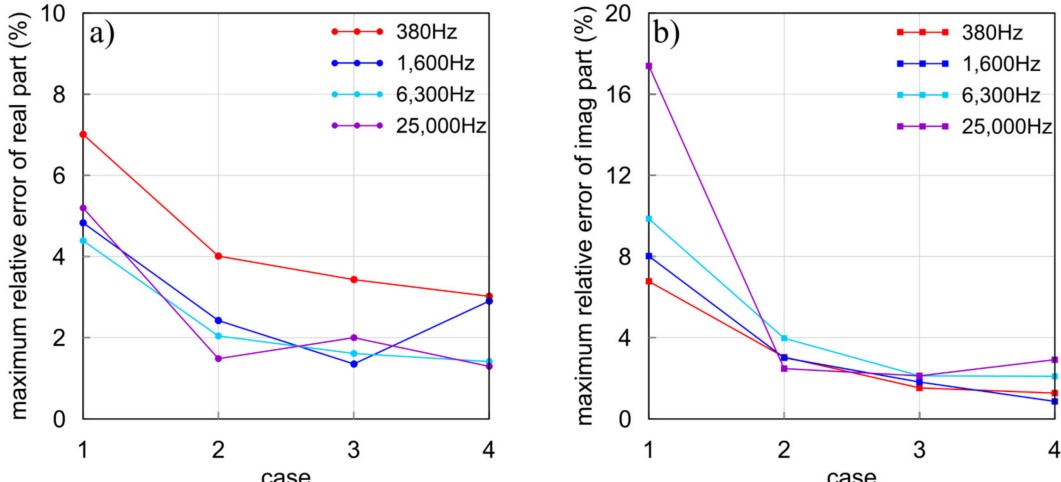

**Figure 11.** The maximum relative errors between WFEM and SEM for Example 1 at different frequencies: (**a**) real part; (**b**) imaginary part.

We then simulate the responses of Example 2 to verify whether our method can handle high resistivity contrast. The model is the same as that in Example 1 except for that the resistivity contrast becomes 0.1 ohm-m for the anomalous body to 500 ohm-m for the half-space. Meanwhile, the same four subdivisions are used to discretize the model and the result from SEM is used as comparison. Figure 12 shows the EM responses, while Table 4 gives the time consumption. From Figures 12 and 13, one can draw similar conclusions to the previous example that with the mesh refinement and scale increasing, the relative errors between our results and the SEM results decrease. The maximum relative error is reduced from 34.8% in Case 1 to about 6% in Case 2 and 3, and to 4.41% in Case 4. Note that the relative error in this example is higher than that in the previous example. This is because when the resistivity contrasts across surfaces become high, the EM field changes sharply. As a result, the accuracy of numerical algorithms is reduced. Even so, our method can still obtain reliable results.

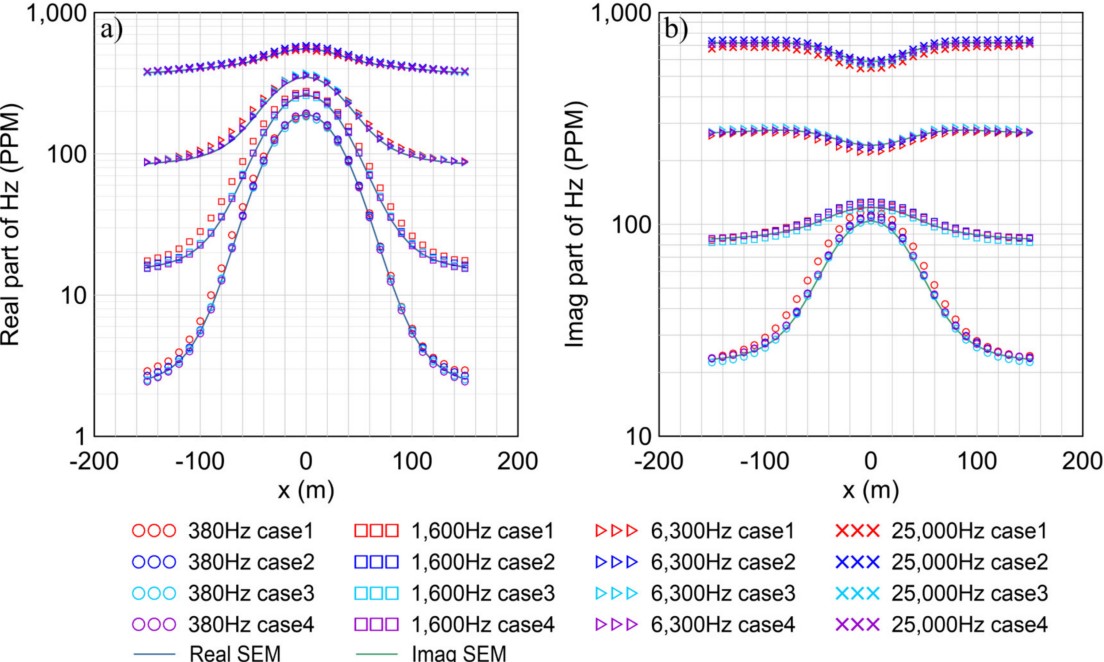

**Figure 12.** EM responses obtained from 4 cases of Example 2 compared to SEM results: (**a**) real part; (**b**) imaginary part.

**Table 4.** Scale of basis function, mesh subdivision, time consumption, and maximum relative error for 4 cases of Example 2.

| Method | Case | Type of Elements | DOFs | Maximum Mesh Size | Minimum Mesh Size | Time Consumption (at 380 Hz) | Maximum Relative Error |
|--------|------|------------------|------|-------------------|-------------------|------------------------------|------------------------|
| WFEM | 1 | $BSWI_{21}$ | 89,900 | 40 m × 40 m × 40 m | 40 m × 40 m × 20 m | 25.360 s | 34.8% |
| - | 2 | $BSWI_{21}$ | 414,540 | 20 m × 20 m × 20 m | 20 m × 20 m × 20 m | 320.319 s | 6.38% |
| - | 3 | $BSWI_{22}$ | 792,756 | 40 m × 40 m × 40 m | 40 m × 40 m × 20 m | 806.477 s | 6.34% |
| - | 4 | $BSWI_{22}$ | 968,436 | 20 m × 40 m × 40 m | 20 m × 20 m × 20 m | 1322.695 s | 4.41% |

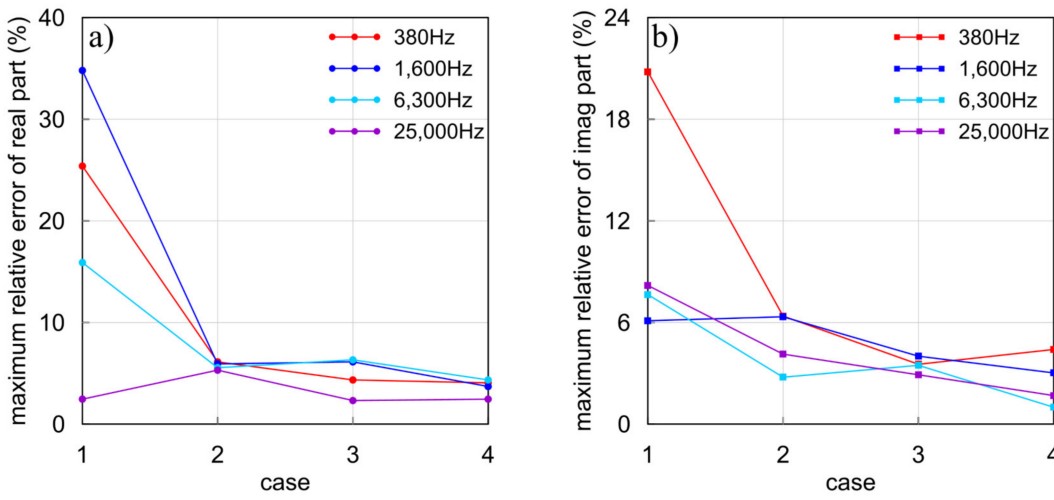

**Figure 13.** The maximum relative errors between WFEM and SEM for Example 2 at different frequencies: (**a**) real part; (**b**) imaginary part.

## 4. Conclusions

In this paper, we have successfully developed the wavelet finite-element method based on the B-spline wavelet on the interval for 3D frequency-domain airborne EM modeling. The BSWI, as the basis function, can better approximate the field variation than the conventional polynomial basis function. Due to special design on the boundary and local support, our method can avoid the Runge phenomenon at high scales, since this method creates a sparser matrix than the traditional FEM method so that the efficiency can be improved. Comparison with 1D semi-analytical solutions validates our method. The numerical experiments for typical models demonstrated that our method has higher accuracy and efficiency than conventional FEM methods. As a high-order numerical method, our algorithm is more flexible in that it is less dependent on the mesh subdivision. Moreover, our method can improve the accuracy by either refining the mesh or increasing the scale of BSWI basis. This obviously offers us more possibilities for accurate numerical modeling and inversions. More specifically, we can use coarse meshes and low-scale BSWI basis functions in the early iterations of the inversion process to recover the rough structure of the underground. With increasing iteration steps, we can use finer meshes and higher-scale BSWI basis to improve the resolution of the inversion continuously, so that we can obtain a steady solution. We must point out that the study in this paper is still preliminary. The structured hexahedral mesh has limitations in fitting the topography or complex underground structures. An unstructured mesh is without doubt the best choice. In addition, the p-type adaptive algorithm that allows local scale enhancement and 3D inversions that adapt to the flexibility of our method are of great attraction to our research. All these will be our future research focus.

**Author Contributions:** Conceptualization, L.G., C.Y. and Y.L.; Formal analysis, L.G., C.Y., X.R., B.Z. and B.X.; Funding acquisition, C.Y. and B.Z.; Investigation, L.G. and N.W.; Methodology, L.G., J.Z. and Y.L.; Software, L.G.; Visualization, L.G. and N.W.; Writing—original draft, L.G.; Writing—review & editing, L.G. and C.Y. All authors have read and agreed to the published version of the manuscript.

**Funding:** This paper is financially supported by the National Natural Science Foundation of China (42030806, 41804098, 42074120, 41904104, 41774125).

**Conflicts of Interest:** The authors declare no conflict of interest.

## Appendix A. The Connection Coefficients

Since the BSWI in Equation (20) has explicit expressions, the calculation of connection coefficients is simple. In this section, we will give the expressions of four types of connection coefficients at different scales. For BSWI$_{21}$ elements, all connection coefficients are $3 \times 3$ matrix that can be written as

$$\mathbf{\Lambda}_1^{0,0} = \mathbf{\Lambda}_2^{0,0} = \mathbf{\Lambda}_3^{0,0} = \frac{1}{12} \begin{pmatrix} 2 & 1 & \\ 1 & 4 & 1 \\ & 1 & 2 \end{pmatrix} \tag{A1}$$

$$\mathbf{\Lambda}_1^{1,1} = \mathbf{\Lambda}_2^{1,1} = \mathbf{\Lambda}_3^{1,1} = \begin{pmatrix} 2 & -2 & \\ -2 & 4 & -2 \\ & -2 & 2 \end{pmatrix} \tag{A2}$$

$$\mathbf{\Lambda}_1^{1,0} = \mathbf{\Lambda}_2^{1,0} = \mathbf{\Lambda}_3^{1,0} = \frac{1}{2} \begin{pmatrix} -1 & -1 & \\ 1 & & -1 \\ & 1 & 1 \end{pmatrix} \tag{A3}$$

$$\mathbf{\Lambda}_1^{0,1} = \mathbf{\Lambda}_2^{0,1} = \mathbf{\Lambda}_3^{0,1} = \frac{1}{2} \begin{pmatrix} -1 & 1 & \\ -1 & & 1 \\ & -1 & 1 \end{pmatrix} \tag{A4}$$

In the case of BSWI$_{22}$ elements, the connection coefficients are all $5 \times 5$ matrix, which are

$$\mathbf{\Lambda}_1^{0,0} = \mathbf{\Lambda}_2^{0,0} = \mathbf{\Lambda}_3^{0,0} = \frac{1}{24} \begin{pmatrix} 2 & 1 & & & \\ 1 & 4 & 1 & & \\ & 1 & 4 & 1 & \\ & & 1 & 4 & 1 \\ & & & 1 & 2 \end{pmatrix} \tag{A5}$$

$$\mathbf{\Lambda}_1^{1,1} = \mathbf{\Lambda}_2^{1,1} = \mathbf{\Lambda}_3^{1,1} = \begin{pmatrix} 4 & -4 & & & \\ -4 & 8 & -4 & & \\ & -4 & 8 & -4 & \\ & & -4 & 8 & -4 \\ & & & -4 & 4 \end{pmatrix} \tag{A6}$$

$$\mathbf{\Lambda}_1^{1,0} = \mathbf{\Lambda}_2^{1,0} = \mathbf{\Lambda}_3^{1,0} = \frac{1}{2} \begin{pmatrix} -1 & 1 & & & \\ -1 & & 1 & & \\ & -1 & & 1 & \\ & & -1 & & 1 \\ & & & -1 & 1 \end{pmatrix} \tag{A7}$$

$$\mathbf{\Lambda}_1^{0,1} = \mathbf{\Lambda}_2^{0,1} = \mathbf{\Lambda}_3^{0,1} = \frac{1}{2} \begin{pmatrix} -1 & -1 & & & \\ 1 & & -1 & & \\ & 1 & & -1 & \\ & & 1 & & -1 \\ & & & 1 & 1 \end{pmatrix}. \tag{A8}$$

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
