# Peer review of "3D Wavelet Finite-Element Modeling of Frequency-Domain Airborne EM Data Based on B-Spline Wavelet on the Interval Using Potentials"

_remotesensing, doi:10.3390/rs13173463_

Round 1

Reviewer 1 Report

The manuscript “3D wavelet finite-element modeling of frequency-domain airborne EM data based on B-spline wavelet on the interval using potentials” describes a quite novel finite-element approach based B-spline wavelet for 3D frequency-domain EM modelling.

The paper is generally well-written, organized, and clear. And it would be definitely of interest to the Readers.

There is clearly a problem with the mathematical symbols formatting. And this makes it not always easy to follow what the Authors mean. Connected to this, there are a few lines that are completely obscure to me.

The quality of Figure 3 must be improved.

Whereas, Figure 10 might be completely removed: my impression is that it is not very informative.

A few additional remarks from my side can be found in the attached pdf.

I hope this is helpful.

Regards

Author Response

Dear RS editors and reviewers:
    Thank you very much for your comments concerning our manuscript "3D wavelet finite-element modeling of frequency-domain airborne EM data based on B-spline wavelet on the interval using potentials" (ID: remotesensing-1331374). Those comments are all very valuable and helpful for improving the clarity of our paper. We have carefully studied them and made corrections accordingly. Enclosed please find the new version where the revised portions are highlighted in the paper. We also like to address some important comments. Please see the attachment.

With best regards,

Changchun Yin

Reviewer 2 Report

The manuscript deals with a method for three-dimensional frequency-domain airborne EM modelling. The principal (and new) aspect is the use of bi-splines wavelets to define the basis function in the finite element method. This is an important improvement, as the author show.

I’d like to congratulate the authors for their work. There is a few things that need attention before acceptance for publications:

-there are need of some references, for example in line 53,

-the author’s name should appears in line 95 related to reference (39),

-I’d like to see a3D example considering a higher resistivity contrast (for example, 0.1/500),

-if possible, the author should show results comparing with other codes (from different authors).

Author Response

(The authors gave the same response as above.)
